| 1<br>2                                                                                                                                         | Title: X-BAND DUAL-POLARIZED RADAR QUANTITATIVE PRECIPITATION ESTIMATE ANALYSES IN THE MIDWESTERN UNITED STATES                                                                                                                                                                                                                                                                                                                                                                                                                                                                                                                                                                                                                                                                                                                                                                                                                                                                                                                                                                                                                                                                                            |
|------------------------------------------------------------------------------------------------------------------------------------------------|------------------------------------------------------------------------------------------------------------------------------------------------------------------------------------------------------------------------------------------------------------------------------------------------------------------------------------------------------------------------------------------------------------------------------------------------------------------------------------------------------------------------------------------------------------------------------------------------------------------------------------------------------------------------------------------------------------------------------------------------------------------------------------------------------------------------------------------------------------------------------------------------------------------------------------------------------------------------------------------------------------------------------------------------------------------------------------------------------------------------------------------------------------------------------------------------------------|
| 3                                                                                                                                              | Authors: Micheal J. Simpson <sup>1</sup> and Neil I. Fox <sup>2</sup>                                                                                                                                                                                                                                                                                                                                                                                                                                                                                                                                                                                                                                                                                                                                                                                                                                                                                                                                                                                                                                                                                                                                      |
| 4                                                                                                                                              |                                                                                                                                                                                                                                                                                                                                                                                                                                                                                                                                                                                                                                                                                                                                                                                                                                                                                                                                                                                                                                                                                                                                                                                                            |
| 5                                                                                                                                              | Micheal J. Simpson <sup>1*</sup>                                                                                                                                                                                                                                                                                                                                                                                                                                                                                                                                                                                                                                                                                                                                                                                                                                                                                                                                                                                                                                                                                                                                                                           |
| 6                                                                                                                                              | Cooperative Institute of Mesoscale Meteorological Studies, University of Oklahoma.                                                                                                                                                                                                                                                                                                                                                                                                                                                                                                                                                                                                                                                                                                                                                                                                                                                                                                                                                                                                                                                                                                                         |
| 7                                                                                                                                              | National Severe Storms Laboratory, Norman, Oklahoma.                                                                                                                                                                                                                                                                                                                                                                                                                                                                                                                                                                                                                                                                                                                                                                                                                                                                                                                                                                                                                                                                                                                                                       |
| 8                                                                                                                                              | Tel: +001 4053256459                                                                                                                                                                                                                                                                                                                                                                                                                                                                                                                                                                                                                                                                                                                                                                                                                                                                                                                                                                                                                                                                                                                                                                                       |
| 9                                                                                                                                              | Email: micheal.simpson@noaa.gov                                                                                                                                                                                                                                                                                                                                                                                                                                                                                                                                                                                                                                                                                                                                                                                                                                                                                                                                                                                                                                                                                                                                                                            |
| 10                                                                                                                                             |                                                                                                                                                                                                                                                                                                                                                                                                                                                                                                                                                                                                                                                                                                                                                                                                                                                                                                                                                                                                                                                                                                                                                                                                            |
| 11                                                                                                                                             | Neil I. Fox <sup>2</sup>                                                                                                                                                                                                                                                                                                                                                                                                                                                                                                                                                                                                                                                                                                                                                                                                                                                                                                                                                                                                                                                                                                                                                                                   |
| 12<br>13                                                                                                                                       | University of Missouri, School of Natural Resources, Water Resources Program, Department of Soil, Environmental, and Atmospheric Sciences, 332 ABNR Building, Columbia, Missouri, USA, 65201.                                                                                                                                                                                                                                                                                                                                                                                                                                                                                                                                                                                                                                                                                                                                                                                                                                                                                                                                                                                                              |
| 14                                                                                                                                             | Tel: +001 5738822144                                                                                                                                                                                                                                                                                                                                                                                                                                                                                                                                                                                                                                                                                                                                                                                                                                                                                                                                                                                                                                                                                                                                                                                       |
| 15                                                                                                                                             | Email: FoxN@Missouri.edu                                                                                                                                                                                                                                                                                                                                                                                                                                                                                                                                                                                                                                                                                                                                                                                                                                                                                                                                                                                                                                                                                                                                                                                   |
| 16                                                                                                                                             |                                                                                                                                                                                                                                                                                                                                                                                                                                                                                                                                                                                                                                                                                                                                                                                                                                                                                                                                                                                                                                                                                                                                                                                                            |
| 17                                                                                                                                             | *Corresponding Author: Micheal J. Simpson (mjs5h7@mail.missouri.edu)                                                                                                                                                                                                                                                                                                                                                                                                                                                                                                                                                                                                                                                                                                                                                                                                                                                                                                                                                                                                                                                                                                                                       |
| 18                                                                                                                                             |                                                                                                                                                                                                                                                                                                                                                                                                                                                                                                                                                                                                                                                                                                                                                                                                                                                                                                                                                                                                                                                                                                                                                                                                            |
|                                                                                                                                                |                                                                                                                                                                                                                                                                                                                                                                                                                                                                                                                                                                                                                                                                                                                                                                                                                                                                                                                                                                                                                                                                                                                                                                                                            |
| 19                                                                                                                                             | Abstract. Over the past decade, polarized weather radars have been at the forefront of the search                                                                                                                                                                                                                                                                                                                                                                                                                                                                                                                                                                                                                                                                                                                                                                                                                                                                                                                                                                                                                                                                                                          |
| 19<br>20                                                                                                                                       | <b>Abstract.</b> Over the past decade, polarized weather radars have been at the forefront of the search for a replacement of estimating precipitation over the spatially, and temporally inferior tipping buckets.                                                                                                                                                                                                                                                                                                                                                                                                                                                                                                                                                                                                                                                                                                                                                                                                                                                                                                                                                                                        |
| 19<br>20<br>21                                                                                                                                 | Abstract. Over the past decade, polarized weather radars have been at the forefront of the search for a replacement of estimating precipitation over the spatially, and temporally inferior tipping buckets.<br>However, many radar-coverage gaps exist within the Continental US (CONUS), proposing a dilemma in                                                                                                                                                                                                                                                                                                                                                                                                                                                                                                                                                                                                                                                                                                                                                                                                                                                                                          |
| 19<br>20<br>21<br>22                                                                                                                           | Abstract. Over the past decade, polarized weather radars have been at the forefront of the search<br>for a replacement of estimating precipitation over the spatially, and temporally inferior tipping buckets.<br>However, many radar-coverage gaps exist within the Continental US (CONUS), proposing a dilemma in<br>that radar rainfall estimate quality degrades with range. One possible solution is that of X-band weather                                                                                                                                                                                                                                                                                                                                                                                                                                                                                                                                                                                                                                                                                                                                                                          |
| 19<br>20<br>21<br>22<br>23                                                                                                                     | Abstract. Over the past decade, polarized weather radars have been at the forefront of the search<br>for a replacement of estimating precipitation over the spatially, and temporally inferior tipping buckets.<br>However, many radar-coverage gaps exist within the Continental US (CONUS), proposing a dilemma in<br>that radar rainfall estimate quality degrades with range. One possible solution is that of X-band weather<br>radars. However, the literature as to their long-term performance is lacking. Therefore, the overarching                                                                                                                                                                                                                                                                                                                                                                                                                                                                                                                                                                                                                                                              |
| 19<br>20<br>21<br>22<br>23<br>24                                                                                                               | Abstract. Over the past decade, polarized weather radars have been at the forefront of the search<br>for a replacement of estimating precipitation over the spatially, and temporally inferior tipping buckets.<br>However, many radar-coverage gaps exist within the Continental US (CONUS), proposing a dilemma in<br>that radar rainfall estimate quality degrades with range. One possible solution is that of X-band weather<br>radars. However, the literature as to their long-term performance is lacking. Therefore, the overarching<br>objective of the current study was to analyze two year's worth of radar data from the X-band dual-                                                                                                                                                                                                                                                                                                                                                                                                                                                                                                                                                        |
| 19<br>20<br>21<br>22<br>23<br>24<br>25                                                                                                         | Abstract. Over the past decade, polarized weather radars have been at the forefront of the search<br>for a replacement of estimating precipitation over the spatially, and temporally inferior tipping buckets.<br>However, many radar-coverage gaps exist within the Continental US (CONUS), proposing a dilemma in<br>that radar rainfall estimate quality degrades with range. One possible solution is that of X-band weather<br>radars. However, the literature as to their long-term performance is lacking. Therefore, the overarching<br>objective of the current study was to analyze two year's worth of radar data from the X-band dual-<br>polarimetric MZZU radar in central Missouri at four separate ranges from the radar, utilizing tipping-                                                                                                                                                                                                                                                                                                                                                                                                                                              |
| <ol> <li>19</li> <li>20</li> <li>21</li> <li>22</li> <li>23</li> <li>24</li> <li>25</li> <li>26</li> </ol>                                     | Abstract. Over the past decade, polarized weather radars have been at the forefront of the search<br>for a replacement of estimating precipitation over the spatially, and temporally inferior tipping buckets.<br>However, many radar-coverage gaps exist within the Continental US (CONUS), proposing a dilemma in<br>that radar rainfall estimate quality degrades with range. One possible solution is that of X-band weather<br>radars. However, the literature as to their long-term performance is lacking. Therefore, the overarching<br>objective of the current study was to analyze two year's worth of radar data from the X-band dual-<br>polarimetric MZZU radar in central Missouri at four separate ranges from the radar, utilizing tipping-<br>buckets as ground-truth precipitation data. The conventional R(Z)-Convective equation, in addition to                                                                                                                                                                                                                                                                                                                                     |
| <ol> <li>19</li> <li>20</li> <li>21</li> <li>22</li> <li>23</li> <li>24</li> <li>25</li> <li>26</li> <li>27</li> </ol>                         | Abstract. Over the past decade, polarized weather radars have been at the forefront of the search<br>for a replacement of estimating precipitation over the spatially, and temporally inferior tipping buckets.<br>However, many radar-coverage gaps exist within the Continental US (CONUS), proposing a dilemma in<br>that radar rainfall estimate quality degrades with range. One possible solution is that of X-band weather<br>radars. However, the literature as to their long-term performance is lacking. Therefore, the overarching<br>objective of the current study was to analyze two year's worth of radar data from the X-band dual-<br>polarimetric MZZU radar in central Missouri at four separate ranges from the radar, utilizing tipping-<br>buckets as ground-truth precipitation data. The conventional R(Z)-Convective equation, in addition to<br>several other polarized algorithms, consisting of some combinations of reflectivity (Z), differential                                                                                                                                                                                                                            |
| <ol> <li>19</li> <li>20</li> <li>21</li> <li>22</li> <li>23</li> <li>24</li> <li>25</li> <li>26</li> <li>27</li> <li>28</li> </ol>             | Abstract. Over the past decade, polarized weather radars have been at the forefront of the search<br>for a replacement of estimating precipitation over the spatially, and temporally inferior tipping buckets.<br>However, many radar-coverage gaps exist within the Continental US (CONUS), proposing a dilemma in<br>that radar rainfall estimate quality degrades with range. One possible solution is that of X-band weather<br>radars. However, the literature as to their long-term performance is lacking. Therefore, the overarching<br>objective of the current study was to analyze two year's worth of radar data from the X-band dual-<br>polarimetric MZZU radar in central Missouri at four separate ranges from the radar, utilizing tipping-<br>buckets as ground-truth precipitation data. The conventional R(Z)-Convective equation, in addition to<br>several other polarized algorithms, consisting of some combinations of reflectivity (ZDR), and the specific differential phase shift (KDP) were used to estimate rainfall. Results                                                                                                                                               |
| <ol> <li>19</li> <li>20</li> <li>21</li> <li>22</li> <li>23</li> <li>24</li> <li>25</li> <li>26</li> <li>27</li> <li>28</li> <li>29</li> </ol> | Abstract. Over the past decade, polarized weather radars have been at the forefront of the search<br>for a replacement of estimating precipitation over the spatially, and temporally inferior tipping buckets.<br>However, many radar-coverage gaps exist within the Continental US (CONUS), proposing a dilemma in<br>that radar rainfall estimate quality degrades with range. One possible solution is that of X-band weather<br>radars. However, the literature as to their long-term performance is lacking. Therefore, the overarching<br>objective of the current study was to analyze two year's worth of radar data from the X-band dual-<br>polarimetric MZZU radar in central Missouri at four separate ranges from the radar, utilizing tipping-<br>buckets as ground-truth precipitation data. The conventional R(Z)-Convective equation, in addition to<br>several other polarized algorithms, consisting of some combinations of reflectivity (Z), differential<br>reflectivity (ZDR), and the specific differential phase shift (KDP) were used to estimate rainfall. Results<br>indicated that the performance of the algorithms containing ZDR were superior in terms of the normalized |

- Furthermore, the R(Z,ZDR) and R(ZDR,KDP) algorithms were the only ones which reported NSE values
- below 100%, whereas R(Z) and R(KDP) equations resulted in false precipitation amounts equal to or
- greater than 65% of the total gauge recorded rainfall amounts. The results show promise in the utilization
- of the smaller, more cost-effective X-band radars in terms of quantitative precipitation estimation at
- ranges from 30 to 80 km from the radar.

### 37 Introduction

Since the late 20<sup>th</sup> Century, weather radars have been at the forefront of multiple studies to determine their accuracy in determining precipitation estimation (e.g., Kitchen and Jackson, 1993; Smith et al., 1996; Ryzhkov et al., 2003; Cunha et al., 2013; Simpson et al., 2016). Multiple researchers have reported accurate measurements in radar rainfall estimates when compared to terrestrial-based precipitation gauges (e.g., tipping buckets). This has several important implications for multi-disciplinary fields which rely on highly spatialized and temporal precipitation data, which can be obtained from radar estimates compared to the spatially inferior rain gauges.

Most studies in the US have utilized the National Weather Service (NWS) Next Generation 46 Radar (NEXRAD) system, comprised of Weather Surveillance Radar – 1988 Doppler (WSR-88D) series 47 instruments, operating at S-band (approximately, 10 - 11 cm) wavelength for their analyses. However, the cost of installation and maintenance of these instruments are much larger in comparison to the smaller, 48 lighter-weighted X-band radars, operating at, approximately, 3 cm wavelength (Matrosov 2010). Berne 49 50 and Krajewski (2013) have stated that, primarily due to the sparse coverage of the WSR-88D S-band radar system, smaller, more frequently-placed X-band radars are a viable option for remediating radar 51 52 rainfall errors that have been recorded at large distances (e.g., Smith et al., 1996; Ryzhkov et al., 2005; Simpson et al., 2016). Although long-term NWS studies have been conducted (Haylock et al., 2008; 53

Fairman et al., 2012; Goudenhoofdt and Delobbe 2012, 2016), the literature of multi-year studies of X-

# 55 band weather radars is not as abundant.

Matrosov et al. (2002) conducted a study analyzing 15 separate rainfall events during an eightweek field campaign in Virginia while utilizing the National Oceanic and Atmospheric Administration's 57 58 (NOAA) X-band dual-polarimetric radar. Several radar rainfall algorithms were implemented, including 59 combinations of the equivalent reflectivity  $(Z_e)$ , differential reflectivity (ZDR), and the specific 60 differential phase-shift (KDP), an R(KDP) equation, and two  $R(Z_e)$  relations, over a region with three ground-truthed rain gauges. It was found that the combined polarimetric estimator (i.e., utilization of Ze, 61 62 KDP, and ZDR) resulted in the overall least standard deviation (22%), while the case-tuned  $R(Z_e)$  relation 63 was slightly higher of 23%. It is noted that R(Ze) measurements are derived from a priori knowledge of 64  $Z_{e}$ , KDP, and ZDR values, whereas the combined polarimetric estimator was not, implying the latter is 65 superior for real-time use. However, the performance of the combined polarimetric estimator works best 66 when rain rates exceeded 1.5 mm  $h^{-1}$ , while  $R(Z_e)$  algorithms were superior at lighter rain rates. Results 67 from the R(KDP) algorithm reported an overall negative bias of 6-9% when compared to the gauge data, in addition to a standard deviation of 30%, primarily due to the sensitivity of KDP measurements while 68 69 utilizing X-band radars.

Expanding upon the literature of implemented R(KDP) algorithms through X-band radars, Wang 71 and Chandrasekar (2010) assessed the performance of three separate R(KDP) algorithms from the 72 Collaborative Adaptive Sensing of the Atmosphere (CASA) Engineering Research Center through the use 73 of the distributed collaborative adaptive sensing (DCAS) network. The DCAS network, essentially, 74 implements multiple radar networks within a relatively small spatial extent, all operating at different 75 volume coverage patterns (VCP's) such that high spatiotemporal resolution data is achieved in addition to 76 overall lower beam height over the area of interest (McLaughlin et al., 2009), mitigating effects that have been observed due to rain rate estimations at large ranges (Kitchen and Jackson, 1993; Ryzhkov et al., 77 78 2005; Simpson et al., 2016). The results indicated that through the use of several different R(KDP)

| 79  | algorithms from multiple different radars, radar Quantitative Precipitation Estimates (QPE) can be              |
|-----|-----------------------------------------------------------------------------------------------------------------|
| 80  | improved significantly. Furthermore, the R(KDP) algorithms exhibited similar bias values (between -6            |
| 81  | and 8 %) that were reported by Matrosov (2002) and Matrosov et al. (2010). However, the normalized              |
| 82  | standard error (NSE) values ranged from, approximately, 16 to 54%, indicating that the overall error in         |
| 83  | R(KDP) rain rate estimates were less than half of the total amount of rain observed for the study.              |
| 84  | The overarching objective of the current study is to add to the relatively few articles on X-band               |
| 85  | dual-polarization radar rain rate performance. Authors have proposed (e.g., Berne and Krajewski 2013)           |
| 86  | that the capability of implementing more X-band radars in comparison to the relatively sparse and               |
| 87  | expensive S-band WSR-88D NEXRAD system to enhance precipitation estimation is a viable option                   |
| 88  | (particularly over the inter-mountain West), especially for hydrologic analyses. However, others (e.g.,         |
| 89  | McLaughlin et al., 2009) suggest the sheer number of radars to achieve such a difference in radar rain rate     |
| 90  | estimation is impractical. Further justification for increasing, at least partially, the construction of X-band |
| 91  | weather radars is necessary through analyses of those that already exist.                                       |
| 92  | The current study analyzes two year's of radar data from the newly-installed dual-polarimetric                  |
| 93  | MZZU X-band radar located in Central Missouri. Over 100 different algorithms were implemented to test           |
| 94  | the performance of the radar while utilizing measurements of reflectivity (Z), differential reflectivity        |
| 95  | (ZDR), and the specific differential phase shift (KDP). Rain rates were calculated based on combinations        |
| 96  | of the aforementioned variables and compared to four separate tipping buckets, which served as ground-          |
| 97  | truth. To determine the performance of all algorithms, multiple statistical analyses were conducted,            |
| 98  | including the bias, mean absolute error, and normalized standard error. Additionally, several contingency       |
| 99  | factors were calculated, such as the overall number of hits, misses, false alarms, and correct negatives.       |
| 100 | Lastly, quantitative analyses, including the missed precipitation amount (MPA), false precipitation             |
| 101 | amount (FPA), and overall error were computed to determine the performance of the 108 algorithms.               |
| 102 | Analyses, such as the current study, are important for determining the accuracy and limitations of dual-        |
| 103 | polarimetric radars such that their incorporation into hydrologic models may be correctly assessed (Ogden       |
|     |                                                                                                                 |

- et al., 1997; Vieux, 2004; Vieux et al., 2004; Vieux and Bedient, 2004; Gourley et al., 2010; Cunha et al.,
- 2015). Furthermore, studies analyzing the performance of X-band radars will allow further indications as
- to whether they should be installed in regions devoid of optimal NWS WSR-88D coverage.
- Data and methodology
- Study location and gauge data

The dates analyzed ranged from August 2015 to August 2017 which, when accounting for radar

down time for maintenance and offline issues, yielded 608 days, or 14952 hours for analyses. The current

study was conducted in Boone County, located in Central Missouri (Figure 1), where the MZZU radar is

located at 38.906°N and 92.269°W. Several Missouri Mesonet rain gauges lie within the domain of the

MZZU radar, located in Versailles, Auxvasse, Williamsburg, and Vandalia, MO, located at,

approximately, 75-km, 30-km, 45-km, and 80-km from the radar, respectively.

Missouri is characterized as a continental type of climate, marked by relatively strong seasonality. 117 Furthermore, Missouri is subject to frequent changes in temperature, primarily due to its inland location 118 and its lack of proximity to any large lakes. All of Missouri experiences below-freezing temperatures on a 119 yearly-basis. The TE525 tipping bucket series performs optimally in temperature conditions between 0 120 and 50°C. Albeit no events recorded a daily maximum temperature above 50°C, 72 days in the cool season (e.g., January and February) recorded temperatures below 0°C. However, only 8 days that 121 122 exhibited sub-freezing average daily temperatures registered precipitation. Thus, less than 1% of the 123 entire data might be further unrepresentative of the actual precipitation. For this study, it was assumed 124 since the amount of precipitation recorded by the gauges during these events were below 5 mm in 125 precipitation, no significant errors would affect the overall statistics.

One tip from the fulcrum device registers 0.254 mm of precipitation, which is the minimum
resolution required for statistics to be analyzed between the radar and the tipping gauge. In spite of the

- well-documented literature discussing the errors associated with tipping buckets (e.g., Ciach and
- Krajewski, 1999a, 1999b; Habib and Krajewski 1999; Habib et al., 2001; Ciach 2002), the gauges are
- well-maintained and well-documented in terms of instrumentation failure, clogging, or other
- discrepancies with the devices. Therefore, they are assumed to be valid as ground-truth devices.
- Radar discussion and data

The radar for the current study is part of the Missouri Experimental Project to Stimulate Competitive Research (EPSCoR) program, primarily aimed at enhancing Missouri's modelling capacity of weather and climate on plants and communities at the local, and regional scale. The X-band radar (MZZU) was installed in the Summer of 2015, in which data acquisition became possible by the Fall of 2015 near the South Farm Research Center, located in central Boone County, MO (Figure 1). The instrument is utilized, primarily, for research purposes, but is also quasi-operational. Specifics regarding the radar are detailed in Table 1.

Raw radar data were analyzed using the Weather Decision Support System - Integrated 142 Information (WDSS-II) program (Lakshmanan et al., 2007a) to assess reflectivity (Z) in addition to dual-143 polarized radar variables including differential reflectivity (ZDR) and specific differential phase shift 144 (KDP). Many different quality control techniques (e.g., Lakshmanan et al., 2007b, 2010, 2014) were 145 implemented to the weather radar data processing with WDSS-II. Three other variables were also generated based on a KDP-based smoothing field (Ryzhkov et al., 2003) for reflectivity, differential 146 reflectivity, and specific differential phase: DSMZ, DZDR, and DKDP, respectively. These were 147 analyzed to determine whether the additional KDP-smoothing fields tend to over- or underestimate QPE's 148 149 (Simpson et al., 2016).

All six variables (Z, ZDR, KDP, DSMZ, DZDR, and DKDP) were converted from their native
polar grid to 256 x 256 1 km Cartesian grids, where the lowest radar elevation scans (0.5°) were used to

| 152 | mitigate uncalculated effects from evaporation and wind drift. An average of 5-minute scans were used           |
|-----|-----------------------------------------------------------------------------------------------------------------|
| 153 | for each of the variables, which were aggregated to hourly totals to be compared to the hourly tipping-         |
| 154 | bucket accumulations. In spite of previous reports suggesting 5 minute to hourly aggregates can have            |
| 155 | significant effects on QPE (Fabry et al., 1994), evidence has been presented that overestimation in             |
| 156 | accumulations may not exceed 26% for a pixel size of 1 km (Shucksmith et al., 2011).                            |
| 157 | The latitude and longitude of each of the 15 gauges were matched with the radar pixel that                      |
| 158 | corresponds to the Cartesian grid value of the seven radar variables which were then implemented in rain        |
| 159 | rate calculations. Three single-polarized $R(Z)$ algorithms were tested, including $R(Z)$ -Convective, $R(Z)$ - |
| 160 | Stratiform, and R(Z)-Tropical. The dual-polarized algorithms implemented are based from previous S-             |
| 161 | and X-band research to more closely resemble the sensitivity of the radars on KDP estimates. Although,          |
| 162 | theoretically, the relationship between R and Z for a well calibrated radar as controlled by the drop size      |
| 163 | distribution should be independent of radar wavelength. However, as the phase shift of the wave is a            |
| 164 | function of the ratio of wavelength to drop radius, the R(KDP) relationships are wavelength dependent.          |
| 165 | The five R(Z,ZDR) S-band equations tested by Simpson et al. (2016) were implemented, whereas                    |
| 166 | six, three, and two X-band R(KDP) algorithms were adopted from Matrosov (2010), Wang and                        |
| 167 | Chandrasekar (2010), and Koffi et al. (2014) (Table 2). Additionally, two X-band R(Z,ZDR) and                   |
| 168 | R(ZDR,KDP) algorithms were adopted from Matrosov (2010) and Koffi et al. (2014), respectively. All              |
| 169 | measures of Z, ZDR, and KDP were tested in addition to their KDP-smoothed derivatives, DSMZ,                    |
| 170 | DZDR, and DKDP. A rain rate echo classification variable (RREC) was also computed, which chooses                |
| 171 | whether an R(Z), R(KDP), R(Z,ZDR), or R(ZDR, KDP) algorithm is implemented in estimating rain rates             |
| 172 | based on the radar fields of Z, ZDR, and KDP (Kessinger et al., 2003). This echo classifier will provide        |
| 173 | evidence as to whether a multi-parameter algorithm is superior to the single algorithms.                        |
| 174 | Furthermore, algorithms were grouped based on the variables implemented to estimate rain rates.                 |
| 175 | Collectively, three R(Z) algorithms were tested, R(Z)-Convective, R(Z)-Stratiform, and R(Z)-Tropical, in        |

addition to the DSMZ counterparts. Five separate R(Z,ZDR) equations were also implemented, including

- five R(Z,DZDR), five R(DSMZ,ZDR), and the five R(DSMZ,DZDR) combinations. These 26 equations
- encompass the S-band algorithms to be tested on the X-band radar, to determine how versatile the
- equations are. Conversely, there were eleven R(KDP) X-band algorithms (and, thus, eleven R(DKDP)
- equations), two R(Z,ZDR), and two R(ZDR,KDP) equations in addition to their DSMZ, DZDR, and
- DKDP variables.

#### 182 Statistical, contingency, and quantitative analyses

The results were split between three different categories: statistical, contingency, and quantitative. 184 The three statistics utilized included the bias, mean absolute error (MAE), and normalized standard error 185 (NSE). The NSE was chosen in place of the root-mean-square-error (RMSE) due to the ambiguity of the 186 measure (Willmott and Matsuura, 2005; Jerez et al., 2013). Contingency analyses included the number of 187 hits, misses, and false alarms. Accounting for the quantitative measure of precipitation due to the number 188 of misses and false alarms, the missed precipitation amount (MPA) and false precipitation amount (FPA) 189 were calculated. Additionally, the sum of precipitation is presented to render a long-term performance 190 (i.e., two year's) of the radar in comparison to the ground-truthed gauges.

#### 192 Results and discussion

193From the four separate gauges, Auxvasse was the closest to the radar (approximately, 30 km)

while Vandalia was the furthest (80 km),