# Peer review of "(untitled)"

_Atmospheric Measurement Techniques, 2017_

## Referee Comment (RC1) · Anonymous Referee #1 · 4 Jan 2018

Review of: X-BAND DUAL-POLARIZED RADAR QUANTITATIVE PRECIPITATION ES-TIMATE ANALYSES IN THE MIDWESTERN UNITED STATES Author(s): Micheal J. Simpson and Neil I. Fox Journal: Atmospheric Measurement Techniques (AMT) MS No.: amt-2017-439

General comments:

This study presents a two-year evaluation of available X-band rainfall relationship performance for a Midwestern US location. The paper falls under the AMT scope as a validation manuscript. An objective is to analyze performance to extended ranges. Overall, the manuscript is not recommended for publication at this time. A resubmis-

sion (rather than revision) is recommended. Several points for the authors to consider.

Extended comments:

1) Conclusions are counter to many published Xband studies. An author statement that initially caught my attention was, "This is surprising since ZDR has not been calibrated for the MZZU radar". While newer radars may have built-in controls to mitigate several issues, it is not typical. My experience is that radar quantities (Z, ZDR, KDP) out-of-the-box are usually not suitable for hydrological applications (esp. over a two-year window). This is a dirty little secret of radar rainfall studies (all wavelengths), e.g., the processing effort to achieve their performance – increasingly problematic when others attempt to replicate performance. This is partially why manuscript "data/code availability" sections are of importance for journals such as AMT (is this now required?).

For a single X-band radar, handling data is nontrivial (attenuation corrections, influences from backscatter differential phase). This reviewer is skeptical, indications are that the authors opted for many tools/methods poorly matched to Xband. For example, superior performance for Z, ZDR-based estimators to longer distance (at X-band) – Z, ZDR can be highly problematic, less informative for 'real-world' rainfall estimation owing to detrimental attenuation corrections in rain, calibration uncertainties, complications in hail, nonuniform beam filling, etc. KDP-based algorithms are typically known for unbiased estimates and most accurate for cumulative (areal, total) studies. The study does not report hourly accumulation comparisons (most common way of visualizing rainfall performance), instead opting for many nonstandard comparisons. For 'false alarm' or detection-type concepts, it is unclear whether cross-correlation coefficient filters (or similar) are used to remove contaminants, other biological/insect echoes, etc. These concepts (thresholds) also change at X-band.

This manuscript indicates that "WDSSII" is used. This is an S-band reference – perhaps NOAA X-POL developments have provided modifications, but these are not obvious; older Ryzhkov technical reports are also S-band references, not X-band.

Note, there has been recent effort put into Xband available to the authors – CSU-Chandraseker, NSSL collaborations (Ryzhkov, Germany – uses of specific attenuation-based estimators), available open-source options. Z and ZDR offsets (e.g., modest 1 dBz or 0.1 dB type) alter performance 20%, moreso with larger coefficients associated with ZDR parameters – also may have other 'hot spot' or similar DSD-related influence modifications, etc., that impact shorter wavelength processing (e.g., Gu et al. 2011). It would be helpful for the authors to demonstrate that basic radar quantity estimates are proper, e.g., scatterplots, or dual-polarization self-consistency examples – as well as details on the typical attenuation corrections (radial examples?), associated coefficients and differential phase processing performed.

2) Given the availability of extended gauge networks, others (including NSSL) have considered rainfall performances to longer distances. For Missouri, I would anticipate that other gauge networks (HADS type, e.g., https://hads.ncep.noaa.gov/) are also available. It would seem that this topic (performance to longer ranges) is still a useful, but needs better support. For an effort that does not introduce a new approach, there is an underwhelming number of gauges / comparisons (as compared to studies that benefit from mesonet gauges, iFloods, etc.). Besides, X-band 'gap filling' idea / motivation is usually not suggestive that X-bands would provide estimates to longer distances, but fill-in and outperform S-band radars in 'gaps' in coverage (lower-levels, etc.). Xband radars are typically not expected to provide rainfall beyond 40 km.

3) Red flag: S-band algorithms. Missouri should be climatologically comparable to Oklahoma, Iowa, Colorado, which have many Xband studies to draw from. There is never justification for S-band algorithms at X-band (e.g., KDP is substantially larger, e.g., 3 times, at X-band than at S-band, etc., and Z, ZDR having different and unique shorter-wavelength nonRayleigh implications, etc.). One would also expect vastly different 'matched' R(Z), R(KDP) relation coefficients (e.g., as from disdrometer, etc).

4) Confusion may also be attributed to selection of metrics (multi-year cumulative comparisons can appear correct for incorrect reasons). Providing performance contingent

on rainfall rate intensity or hourly comparisons is preferable for many reasons, e.g., if the parts to the dataset primarily contributing were 'light' rainfall (R < 5 mm/hr, or hourly accumulation < __ mm, etc.), it may be more acceptable/believable that R(Z, ZDR) was outperforming other methods than in the presence of heavier rainfall, etc. For example, it may be fair to expect R(Z,ZDR) should perform better in light rain to closer (or lengthier) distances, provided there was not much precipitation along that path (aka, attenuation in rain to that location). Many dual-polarization methods tend to work optimally in heavier rainfall, etc.

---

## Referee Comment (RC2) · Anonymous Referee #2 · 8 Jan 2018

General comments:

The paper presents a study of a long term data record from an X-band weather radar which would be an interesting and useful contribution to the community were it not for the notable omissions, inconsistencies and lack of detail within the paper. These deficiencies can be summarised as follows:

1. A lack of detail on the processing applied to the data from the X-band radar and any scientific discussion of how this processing, or omission of, could be affecting the results. The most striking statement in this regard is the following "This is surprising since ZDR has not been calibrated for the MZZU radar" (page 10) which is then not

[Figure]

followed up in any relevant way, such as a discussion of whether an adjustment of the calibration is necessary given the data available. Other issues to consider which are not mentioned but highly relevant are the possibility of reflectivity miscalibration, partial beam blockage, attenuation and differential attenuation correction and the calculation of specific differential phase. There have been several studies recently which covered many of these issues for X-band radar, for example Park et al (2005), Giangrande et al (2014) and Diederich et al (2015).

2. There is also a lack of detail regarding the scanning strategy applied by the radar and any possible impact of this on accuracy. The authors state only 0.5 degree elevation scans are used but how frequently do these occur? Are they regularly spaced or does the scan change throughout the two year period? Is the rotation speed of the radar constant or changeable and what is the rotation speed? Each of these factors will impact on the accuracy of the QPE obtained from the radar and should be covered in the paper.

3. There is a lack of ground observations for verification and an assumption that 4 gauges at different ranges can characterise range effect without considering azimuthal differences or random variation. The paper would benefit greatly from the addition of more ground observations, for example Diederich et al. (2015) use 133 gauges and Giangrande et al. 2014 use 34 sites for their respective studies.

4. It is difficult to follow which algorithms have been applied (108 in total, page 4 or 68, page 10). What value is there from presenting so many, most of which are not described in any detail in the text? The paper would be better if a smaller, more focused selection of algorithms were applied, and these would be best presented in a table similar to table 2 to allow them to be compared easily.

5. The discussion and conclusions are insubstantial. Having compared so many algorithms the discussion on page 16 is lacking. Is there a reason for the algorithms containing reflectivity always having a negative bias, such as miscalibration or beam

blockage? Similarly what could be the cause of the persistent positive bias when using KDP? Again the R(Z,ZDR) and R(ZDR,KDP) algorithms "performed the best" yet have the lowest overall correlation coefficients but there is no consideration of why this should be the case. Looking at figure 6 it is clear that all the methods shown underestimate higher rainfall accumulations yet this is not covered in the text at all.

Recommendation

Though the general idea of the paper has potential the paper requires significant improvement and rewriting before it is suitable for publication. I would recommend the authors focus more on a reduced number of carefully selected algorithms with more discussion on the relative merits of each of them while providing more detail on the processing of the data and the limitations inherent within. If possible a greater number of rain gauges should be used to allow more robust findings to be demonstrated, particularly if the authors wish to concentrate on the effect of range both on the suitability of different algorithms and the overall accuracy of X-band radar QPE.

Diederich, Malte, Alexander Ryzhkov, Clemens Simmer, Pengfei Zhang, and Silke Trömel. 2014. "Use of Specific Attenuation for Rainfall Measurement at X-Band Radar Wavelengths. Part II: Rainfall Estimates and Comparison with Rain Gauges." Journal of Hydrometeorology 16 (2):503–16. https://doi.org/10.1175/JHM-D-14-0067.1.

Giangrande, Scott E., Scott Collis, Adam K. Theisen, and Ali Tokay. 2014. "Precipitation Estimation from the ARM Distributed Radar Network during the MC3E Campaign." Journal of Applied Meteorology and Climatology 53 (9):2130–47. https://doi.org/10.1175/JAMC-D-13-0321.1.

Park, S-G., M. Maki, K. Iwanami, V. N. Bringi, and V. Chandrasekar. 2005. "Correction of Radar Reflectivity and Differential Reflectivity for Rain Attenuation at X Band. Part II: Evaluation and Application." Journal of Atmospheric and Oceanic Technology 22 (11):1633–55. https://doi.org/10.1175/JTECH1804.1.

---

## Referee Comment (RC3) · Anonymous Referee #3 · 18 Jan 2018

Manuscript Review Comments to amt-2017-439

Title: X-BAND DUAL-POLARIZED RADAR QUANTITATIVE PRECIPITATION ESTIMATE ANALYSES IN THE MIDWESTERN UNITED STATES

General Comments: This manuscript evaluates a large number of dual-polarization radar rainfall relations for an X-band radar deployed in central Missouri, USA. Rain gauge data collected during August 2015 to August 2017 are used for quantitative evaluation purposes. Overall, this topic well fits the scope of AMT. However, the manuscript is not well presented. Many fundamental issues in X-band QPE are missing. Following are some of my major concerns and minor comments. In addition, there are small typos

here and there but since I am recommending a rather substantial revision, those issues can be left for a later time. The authors are encouraged to re-submit this manuscript after addressing the following issues.

Major Concerns:

1. Technically, I do not see anything novel in this work. Most of the sciences and principles have already been published in previous studies. Some of the analysis procedures are very similar to what has been used before. However, this manuscript reads like there are not many X-band studies in the literature, which is awkward. The introduction is very roughly written, without referring to proper previous studies. The uniqueness of this manuscript might be its study domain. Unfortunately, the authors fails to elaborate this point.

References:

Anagnostou, M. N., E. N. Anagnostou, and J. Vivekanandan, 2007: Comparison of raindrop size distribution estimates from X-band and S-band polarimetric observations. IEEE Geosci. Remote Sens. Lett., 4, 601-605.

Chandrasekar, V., Y. Wang, and H. Chen, 2012: The CASA quantitative precipitation estimation system: a five year validation study, Natural Hazards and Earth System Sciences, 12, 2811-2820.

Chandrasekar, V., H. Chen, and B. Philips, 2018: Principles of high-resolution radar network for hazard mitigation and disaster management in an urban environment. J. Meteor. Soc. Japan, 96A, https://doi.org/10.2151/jmsj.2018-015.

Chen, H., Lim, S., Chandrasekar, V., and Jang, B.-J., 2017: Urban Hydrological Applications of Dual-Polarization X-Band Radar: Case Study in Korea, Journal of Hydrologic Engineering, 22, E5016001, 10.1061/(asce)he.1943-5584.0001421.

Cifelli, R., V. Chandrasekar, H. Chen, and L. E. Johnson, 2018: High resolution radar quantitative precipitation estimation in the San Francisco Bay Area:

Rainfall monitoring for the urban environment. J. Meteor. Soc. Japan, 96A, https://doi.org/10.2151/jmsj.2018-016.

Kalogiros, J., M. N. Anagnostou, E. N. Anagnostou, M. Montopoli, E. Picciotti, and F. S. Marzano, 2014: Evaluation of a new Polarimetric Algorithm for Rain-Path Attenuation Correction of X-Band Radar Observations Against Disdrometer Data. IEEE Geoscience and Remote Sensing Letters, 52, 1369-1380.

Marzano, F. S., G. Botta, and M. Montopoli, 2010: Iterative Bayesian retrieval of hydrometeor content from X-band polarimetric weather radar. IEEE Trans. Geosci. Remote Sens., 48, 3059-3074.

Matrosov, S. Y., D. E. Kingsmill, B. E. Martner, and F. M. Ralph, 2005: The utility of X-band polarimetric radar for quantitative estimates of rainfall parameters. J. Hydrometeor., 6, 248-262.

Shakti, P. C., M. Maki, S. Shimizu, T. Maesaka, D.-S. Kim, D.-I. Lee, and H. Iida, 2013: Correction of Reflectivity in the Presence of Partial Beam Blockage over a Mountainous Region Using X-Band Dual Polarization Radar. J. Hydrometeor., 14, 744-764.

Shi, Z., H. Chen, V. Chandrasekar, and J. He, 2018: Deployment and Performance of an X-Band Dual-Polarization Radar during the Southern China Monsoon Rainfall Experiment. Atmosphere, 9(1), 4, doi:10.3390/atmos9010004.

2. Details about X-band radar data quality control are NOT enough. In addition, Kdp estimation and attenuation correction are completely neglected. These are all key aspects for X-band QPE. After reading this manuscript, the readers are even sure if the X-band radar data quality is enough for quantitative applications.

3. The authors included a huge number of rainfall relations in the evaluation without explaining why. Many of the relations are wrong (if applied at X-band). Why do you need so many R-Kdp relations? Why are you even implementing S-band R-Kdp relations? Which relation was derived using local raindrop size distribution data?

[Figure]

Other Comments:

4. Page 1, Line 23: However, the literature as to their long-term performance is lacking.

This is not true. Please rephrase the writing. Also refer to the references listed in Major Concern #1.

5. Page 4, Lines 39-40: Most of the references are not current. A few widely used dual-pol rainfall algorithms are suggested here.

References:

Chen, H., V. Chandrasekar, and R. Bechini, 2017: An Improved Dual-Polarization Radar Rainfall Algorithm (DROPS2.0): Application in NASA IFloodS Field Campaign. J. Hydrometeor., 18, 917-937.

Cifelli, R., V. Chandrasekar, S. Lim, P. C. Kennedy, Y. Wang, and S. A. Rutledge, 2011: A new dual-polarization radar rainfall algorithm: Application in Colorado precipitation events. J. Atmos. Oceanic Technol., 28, 352-364.

Giangrande, S.E., and Ryzhkov, A.V., 2008: Estimation of rainfall based on the results of polarimetric echo classification. J. Appl. Meteorol. Climate, 47, 2445-2462.

Ryzhkov, A.V., T.J. Schuur, D.W. Burgess, P.L. Heinselman, S.E. Giangrande, and D.S. Zrnic, 2005: The Joint Polarization Experiment: Polarimetric Rainfall Measurements and Hydrometeor Classification. Bull. Amer. Meteor. Soc., 86, 809-824.

6. Page 2, Line 48-49: However, the cost . . .are much larger. . ..

Please use proper reference (i.e., McLaughlin et al. 2009).

Reference:

McLaughlin, D., D. Pepyne, B. Philips, and Coauthors, 2009: Short-Wavelength Technology and the Potential For Distributed Networks of Small Radar Systems. Bull. Amer. Meteor. Soc., 90, 1797-1817.

7. Page 4, Line 82: ...normalized standard error (NSE)...

NSE has been expanded too many times all through this manuscript. Please pay attention to the usage of acronyms.

8. Page 4, Line 84: ... relatively few articles on X-band...

There are many X-band QPE studies. Please rephrase the writing. Again, the uniqueness of this study should be emphasized (particularly the local precipitation microphysics).

9. Page 4, Line 93: Over 100 different algorithms were implemented...

This is very strange! Do you really need 100+ algorithms? The authors should pay more attention to the algorithms that can better reflect local rainfall microphysics. Most of the relations are taken from other papers that focused on different regions. Some of the algorithms used in this study were not even designed for X-band... A detailed investigation of local precipitation microphysics will be helpful.

10. Page 5, Lines 105-106: ...X-band radars will allow further indications as to whether they should be installed in regions devoid of optimal NWS WSR-88D coverage.

The description on such aspect is weak. The authors may want to rephrase this sentence or refer to other studies.

11. Page 7, Lines 162-163: R and Z ... should be independent of radar wavelength.

This is true only when you are assuming Rayleigh scattering. Please clarify!

12. Page 9, Line 210: $p < 0.10$

What is p? Probability?

13. Page 23, Figure 2. Please include corresponding rainfall products derived from this radar.

---

## Author Comment (AC1) · 3 Mar 2018

Review of: X-BANDDUAL-POLARIZEDRADARQUANTITATIVEPRECIPITATIONESTIMATE ANALYSES IN THE MIDWESTERN UNITED STATES Author(s): Micheal J. Simpson and Neil I. Fox Journal: Atmospheric Measurement Techniques (AMT) MS No.: amt-2017-439 General comments: This study presents a two-year evaluation of available X-band rainfall relationship performance for a Midwestern US location. The paper falls under the AMT scope as a validation manuscript. An objective is to analyze performance to extended ranges. Overall, the manuscript is not recommended for publication at this time. A resubmission (rather than revision) is recommended. Several points for the authors to consider. Extended comments:

1 ) Conclusions are counter to many published Xband studies. An author statement that initially caught my attention was, "This is surprising since ZDR has not been calibrated for the MZZU radar". While newer radars may have built-in controls to mitigate several issues, it is not typical. My experience is that radar quantities(Z,ZDR,KDP) out-of-the box are usually not suitable for hydrological applications (esp. over a two-year window). This is a dirty little secret of radar rainfall studies (all wavelengths), e.g., the processing effort to achieve their performance – increasingly problematic when others attempt to replicate performance. This is partially why manuscript "data/code availability" sections are of importance for journals such as AMT (is this now required?).

We appreciate this reviewer comment. We have revised this wording in our document to describe that although ZDR has not been calibrated, some of the errors may seem to offset over the long-term, revealing issues with long-term studies as opposed to short-term analyses. We are aware that it is ideal for, at least, S-band radars to fall within +/- 0.2 dB for accurate R(Z,ZDR) or R(ZDR,KDP) estimates. However, the system ZDR (ZDR offset) data, receiver/transmitter/sun biases were not available for analyses, but will be for future studies.

For a single X-band radar, handling data is nontrivial (attenuation corrections, influences from backscatter differential phase). This reviewer is skeptical, indications are that the authors opted for many tools/methods poorly matched to Xband. For example, superior performance for Z, ZDR-based estimators to longer distance (at X-band) – Z, ZDR can be highly problematic, less informative for 'real-world' rainfall estimation owing to detrimental attenuation corrections in rain, calibration uncertainties, complications in hail, nonuniform beam filling, etc. KDP-based algorithms are typically known for unbiased estimates and most accurate for cumulative (areal, total) studies. The study does not report hourly accumulation comparisons (most common way of visualizing rainfall performance), instead opting for many nonstandard comparisons. For 'false alarm' or detection-type concepts, it is unclear whether cross-correlation coefficient filters (or similar) are used to remove contaminants, other biological/insect echoes, etc. These concepts (thresholds) also change at X-band).

This manuscript indicates that "WDSSII" is used. This is an S-band reference – perhaps NOAA X-POL developments have provided modifications, but these are not obvious; older Ryzhkov technical reports are also S-band references, not X-band. Note, there has been recent effort put into Xband available to the authors – CSU Chandraseker, NSSL collaborations (Ryzhkov, Germany–use of specific attenuation based estimators), available open-source options. Z and ZDR offsets (e.g., modest 1 dBz or 0.1 dB type) alter performance 20%, moreso with larger coefficients associated with ZDR parameters – also may have other 'hot spot' or similar DSD-related influence modifications, etc., that impact shorter wavelength processing (e.g., Gu et al. 2011). It would be helpful for the authors to demonstrate that basic radar

quantity estimates are proper, e.g., scatterplots, or dual-polarization self-consistency examples – as well as details on the typical attenuation corrections (radial examples?), associated coefficients and differential phase processing performed.

We thank the reviewer for these comments. To the best of the authors' knowledge, there is no direct algorithm in WDSSII that can specifically handle X-band data. However, the raw data and slightly QC'd data have been handled from the OU's mobile X-band weather radar and showed promising results in removing biological filters, ground clutter, and sun spikes. Regardless, this does not solve the issue of overall calibration previously mentioned, but is surprising the performance of particular algorithms for QPE.

2) Given the availability of extended gauge networks, others (including NSSL) have considered rainfall performances to longer distances. For Missouri, I would anticipate that other gauge networks (HADS type, e.g., https://hads.ncep.noaa.gov/) are also available. It would seem that this topic (performance to longer ranges) is still a useful, but needs better support. For an effort that does not introduce a new approach, there is an underwhelming number of gauges / comparisons (as compared to studies that benefit from mesonet gauges, iFloods, etc.). Besides, X-band 'gap filling' idea / motivation is usually not suggestive that X-bands would provide estimates to longer distances, but fill-in and outperform S-band radars in 'gaps' in coverage (lower-levels, etc.). Xband radars are typically not expected to provide rainfall beyond 40 km.

One of the surprising findings from the study is that the performance of the algorithms did not degrade as quickly as one would think; NSE's in the region of 100% beyond 40 km were still possible for specific algorithms in spite of the lack of calibration, leading to a further justification for using X-band radars for 'gap-filling' purposes.

Although HADS, CoCoRaHS, GHCN, USHCN, etc. have gauges for Missouri, they tend to be sparsely available for the center of the state. We have gathered data from many of these networks and will provide a more in-depth analyses of fewer algorithms with more gauges to prove the robust capabilities of the algorithms implemented. However, for many regions (particularly to the West), gauges are lacking to validate QPE, which was demonstrated for this paper.

3) Red flag: S-band algorithms. Missouri should be climatologically comparable to Oklahoma, Iowa, Colorado, which have many Xband studies to draw from. There is never justification for S-band algorithms at X-band (e.g., KDP is substantially larger, e.g., 3 times, at X-band than at S-band, etc., and Z, ZDR having different and unique shorter-wavelength nonRayleigh implications, etc.). One would also expect vastly different 'matched' R(Z), R(KDP) relation coefficients (e.g., as from disdrometer, etc).

The inclusion of the S-band algorithms proved that the KDP-containing algorithms did, indeed, provide vastly larger QPE's, yet for algorithms containing only Z or ZDR, they performed very well. Assuming standard Rayleigh-scattering, the Z-containing algorithms should, theoretically, not be effected as much as the KDP algorithms, which was demonstrated in this paper.

4) Confusion may also be attributed to selection of metrics (multi-year cumulative comparisons can appear correct for incorrect reasons). Providing performance contingent on rainfall rate intensity or hourly comparisons is preferable for many reasons, e.g., if the parts to the dataset primarily contributing were 'light' rainfall (R < 5 mm/hr, or hourly accumulation < ___ mm, etc.), it may be more

acceptable/believable that R(Z, ZDR) was outperforming other methods than in the presence of heavier rainfall, etc. For example, it may be fair to expect R(Z,ZDR) should perform better in light rain to closer (or lengthier) distances, provided there was not much precipitation along that path (aka, attenuation in rain to that location). Many dual-polarization methods tend to work optimally in heavier rainfall, etc.

Thank you for this comment. The inclusion of statistical analyses with respect to rain rate would be a very good idea. This would decipher which algorithm performs better in different seasons and whether stratiform/convective events were more prevalent. Currently, a radome-wetting algorithm is underway to mitigate the effects of heavy rainfall directly over the radar (due to a complete loss in signal).

---

## Author Comment (AC2) · 3 Mar 2018

General comments: The paper presents a study of a long term data record from an X-band weather radar which would be an interesting and useful contribution to the community were it not for the notable omissions, inconsistencies and lack of detail within the paper. These deficiencies can be summarised as follows:

1. A lack of detail on the processing applied to the data from the X-band radar and any scientific discussion of how this processing, or omission of, could be affecting the results. The most striking statement in this regard is the following "This is surprising since ZDR has not been calibrated for the MZZU radar" (page 10) which is then not followed up in any relevant way, such as a discussion of whether an adjustment of the calibration is necessary given the data available. Other issues to consider which are not mentioned but highly relevant are the possibility of reflectivity miscalibration, partial beam blockage, attenuation and differential attenuation correction and the calculation of specific differential phase. There have been several studies recently which covered many of these issues for X-band radar, for example Park et al (2005), Giangrande et al (2014) and Diederich et al (2015).

We appreciate this reviewer comment. We have revised this wording in our document to describe that although ZDR has not been calibrated, some of the errors may seem to offset over the long-term, revealing issues with long-term studies as opposed to short-term analyses. We are aware that it is ideal for, at least, S-band radars to fall within +/- 0.2 dB for accurate R(Z,ZDR) or R(ZDR,KDP) estimates. However, the system ZDR (ZDR offset) data, receiver/transmitter/sun biases were not available for analyses, but will be for future studies. Furthermore, previous studies by the authors have shown that over long-term, misses/false alarms do, indeed, tend to offset to perceive the data as "good" whereas, in reality, they may have large, long-term errors. This prompted the usage of the FPA and MPA to determine these errors, which were both still relatively small given the time period.

2. There is also a lack of detail regarding the scanning strategy applied by the radar and any possible impact of this on accuracy. The authors state only 0.5 degree elevation scans are used but how frequently do these occur? Are they regularly spaced or does the scan change throughout the two year period? Is the rotation speed of the radar constant or changeable and what is the rotation speed? Each of these factors will impact on the accuracy of the QPE obtained from the radar and should be covered in the paper.

Thank you for this comment. We have provided information as to the scanning strategy as well as the typical scanning elevation.

3. There is a lack of ground observations for verification and an assumption that 4 gauges at different ranges can characterise range effect without considering azimuthal differences or random variation. The paper would benefit greatly from the addition of more ground observations, for example Diederich et al. (2015) use 133 gauges and Giangrande et al. 2014 use 34 sites for their respective studies.

Thank you for this comment. Although HADS, CoCoRaHS, GHCN, USHCN, etc. have gauges for Missouri, they tend to be sparsely available for the center of the state. We have gathered data from many of these networks and will provide a more in-depth analyses of fewer algorithms with more gauges to

prove the robust capabilities of the algorithms implemented. However, for many regions (particularly to the West), gauges are lacking to validate QPE, which was demonstrated for this paper.

4. It is difficult to follow which algorithms have been applied (108 in total, page 4 or 68, page 10). What value is there from presenting so many, most of which are not described in any detail in the text? The paper would be better if a smaller, more focused selection of algorithms were applied, and these would be best presented in a table similar to table 2 to allow them to be compared easily.

We appreciate this comment, and look forward to providing a more in-depth analysis of the best-performing algorithms over a broader-range of ground-truthed rainfall sources. We are also getting ready to implement disdrometer data to help with calibration and, hopefully, derive our own equations.

5. The discussion and conclusions are insubstantial. Having compared so many algorithms the discussion on page 16 is lacking. Is there a reason for the algorithms containing reflectivity always having a negative bias, such as miscalibration or beam blockage? Similarly what could be the cause of the persistent positive bias when using KDP? Again the R(Z,ZDR) and R(ZDR,KDP) algorithms "performed the best" yet have the lowest overall correlation coefficients but there is no consideration of why this should be the case. Looking at figure 6 it is clear that all the methods shown underestimate higher rainfall accumulations yet this is not covered in the text at all. Recommendation Though the general idea of the paper has potential the paper requires significant improvement and rewriting before it is suitable for publication. I would recommend the authors focus more on a reduced number of carefully selected algorithms with more discussion on the relative merits of each of them while providing more detail on the processing of the data and the limitations inherent within. If possible a greater number of rain gauges should be used to allow more robust findings to be demonstrated, particularly if the authors wish to concentrate on the effect of range both on the suitability of different algorithms and the overall accuracy of X-band radar QPE.

Diederich, Malte, Alexander Ryzhkov, Clemens Simmer, Pengfei Zhang, and Silke Trömel. 2014. "Use of Specific Attenuation for Rainfall Measurement at X-Band Radar Wavelengths. Part II: Rainfall Estimates and Comparison with Rain Gauges." Journal of Hydrometeorology 16 (2):503–16. https://doi.org/10.1175/JHM-D-14-0067.1.

Giangrande, Scott E., Scott Collis, Adam K. Theisen, and Ali Tokay. 2014. "Precipitation Estimation from the ARM Distributed Radar Network during the MC3E Campaign." Journal of Applied Meteorology and Climatology 53 (9):2130–47. https://doi.org/10.1175/JAMC-D-13-0321.1.

Park, S-G., M. Maki, K. Iwanami, V. N. Bringi, and V. Chandrasekar. 2005. "Correction of Radar Reflectivity and Differential Reflectivity for Rain Attenuation at X Band. Part II: Evaluation and Application." Journal of Atmospheric and Oceanic Technology 22 (11):1633–55. https://doi.org/10.1175/JTECH1804.1.

AMTD

---

## Author Comment (AC3) · 3 Mar 2018

Manuscript Review Comments to amt-2017-439 Title: X-BAND DUAL-POLARIZED RADAR QUANTITATIVE PRECIPITATION ESTIMATE ANALYSES IN THE MIDWESTERN UNITED STATES General Comments: This manuscript evaluates a large number of dual-polarization radar rainfall relations for an X-band radar deployed in central Missouri, USA. Rain gauge data collected during August 2015 to August2017 are used for quantitative evaluation purposes. Overall, this topic well fits the scope of AMT. However, the manuscript is not well presented. Many fundamental issues in X-band QPE are missing. Following are some of my major concerns and minor comments. In addition, there are small typos here and there but since I am recommending a rather substantial revision, those issues can be left for a later time. The authors are encouraged to re-submit this manuscript after addressing the following issues.

Major Concerns: 1. Technically, I do not see anything novel in this work. Most of the sciences and principles have already been published in previous studies. Some of the analysis procedures are very similar to what has been used before. However, this manuscript reads like there are not many X-band studies in the literature, which is awkward. The introduction is very roughly written, without referring to proper previous studies. The uniqueness of this manuscript might be its study domain. Unfortunately, the authors fails to elaborate this point.

We thank the reviewer for this comment, and we have added proper, more recent literature.

References: Anagnostou, M. N., E. N. Anagnostou, and J. Vivekanandan, 2007: Comparison of raindrop size distribution estimates from X-band and S-band polarimetric observations. IEEE Geosci. Remote Sens. Lett., 4, 601-605.

Chandrasekar, V., Y. Wang, and H. Chen, 2012: The CASA quantitative precipitation estimation system: a five year validation study, Natural Hazards and Earth System Sciences, 12, 2811-2820.

Chandrasekar, V., H. Chen, and B. Philips, 2018: Principles of high-resolution radar network for hazard mitigation and disaster management in an urban environment. J. Meteor. Soc. Japan, 96A, https://doi.org/10.2151/jmsj.2018-015.

Chen, H., Lim, S., Chandrasekar, V., and Jang, B.-J., 2017: Urban Hydrological Applications of Dual-Polarization X-Band Radar: Case Study in Korea, Journal of Hydrologic Engineering, 22, E5016001, 10.1061/(asce)he.1943-5584.0001421.

 Cifelli, R., V. Chandrasekar, H. Chen, and L. E. Johnson, 2018: High resolution radar quantitative precipitation estimation in the San Francisco Bay Area: Rainfall monitoring for the urban environment. J. Meteor. Soc. Japan, 96A, https://doi.org/10.2151/jmsj.2018-016.

Kalogiros, J., M. N. Anagnostou, E. N. Anagnostou, M. Montopoli, E. Picciotti, and F. S. Marzano, 2014: Evaluation of a new Polarimetric Algorithm for Rain-Path Attenuation Correction of X-Band Radar Observations Against Disdrometer Data. IEEE Geoscience and Remote Sensing Letters, 52, 1369-1380.

Marzano, F. S., G. Botta, and M. Montopoli, 2010: Iterative Bayesian retrieval of hydrometeor content from X-band polarimetric weather radar. IEEE Trans. Geosci. Remote Sens., 48, 3059-3074.

Matrosov, S. Y., D. E. Kingsmill, B. E. Martner, and F. M. Ralph, 2005: The utility of X-band polarimetric radar for quantitative estimates of rainfall parameters. J. Hydrometeor., 6, 248-262.

Shakti, P. C., M. Maki, S. Shimizu, T. Maesaka, D.-S. Kim, D.-I. Lee, and H. Iida, 2013: Correction of Reflectivity in the Presence of Partial Beam Blockage over a Mountainous Region Using X-Band Dual Polarization Radar. J. Hydrometeor., 14, 744-764.

Shi, Z., H. Chen, V. Chandrasekar, and J. He, 2018: Deployment and Performance of an X-Band Dual-Polarization Radar during the Southern China Monsoon Rainfall Experiment. Atmosphere, 9(1), 4, doi:10.3390/atmos9010004.

2. Details about X-band radar data quality control are NOT enough. In addition, Kdp estimation and attenuation correction are completely neglected. These are all key aspects for X-band QPE. After reading this manuscript, the readers are even sure if the X-band radar data quality is enough for quantitative applications.

We appreciate this comment. We address the fact the radar is uncalibrated, and little-to-no quality control methods have been conducted. Still, we present rather promising results, which could only be further improved by adding these methods in our future QPE methods (which will be presented with larger gauge networks).

3. The authors included a huge number of rainfall relations in the evaluation without explaining why. Many of the relations are wrong (if applied at X-band). Why do you need so many R-Kdp relations? Why are you even implementing S-band R-Kdp relations? Which relation was derived using local raindrop size distribution data?

Thank you for this comment. We tested many different algorithm to test if they were robust to be transferred to ther X-band radars where disdrometic data is unavailable (such as for the current study). We implemented S-band algorithms such that the Z-containing algorithms proved do rather well in QPE, while the S-band KDP algorithms, indeed, showed very high QPE's. This study is just validating the large rainrates if S-band R(KDP) equations are used as opposed to X-band.

4. Page1,Line23: However, the literature as to their long-term performance is lacking. This is not true. Please rephrase the writing. Also refer to the references listed in Major Concern #1.

We appreciate this comment and have made the necessary changes.

5. Page 4, Lines 39-40: Most of the references are not current. A few widely used dual-pol rainfall algorithms are suggested here. References:

We thank the reviewer for this comment and have added the references accordingly.

Chen, H., V. Chandrasekar, and R. Bechini, 2017: An Improved Dual-Polarization Radar Rainfall Algorithm (DROPS2.0): Application in NASA IFloodS Field Campaign. J. Hydrometeor., 18, 917-937.

Cifelli, R., V. Chandrasekar, S. Lim, P. C. Kennedy, Y. Wang, and S. A. Rutledge, 2011: A new dual-polarization radar rainfall algorithm: Application in Colorado precipitation events. J. Atmos. Oceanic Technol., 28, 352-364.

Giangrande, S.E., and Ryzhkov, A.V., 2008: Estimation of rainfall based on the results of polarimetric echo classification. J. Appl. Meteorol. Climate, 47, 2445-2462.

Ryzhkov, A.V., T.J. Schuur, D.W. Burgess, P.L. Heinselman, S.E. Giangrande, and D.S. Zrnic, 2005: The Joint Polarization Experiment: Polarimetric Rainfall Measurements and Hydrometeor Classification. Bull. Amer. Meteor. Soc., 86, 809-824.

6. Page 2, Line 48-49: However, the cost ...are much larger.... Please use proper reference (i.e., McLaughlin et al. 2009).

The proper reference has been added. We thank the reviewer for this comment.

Reference: McLaughlin, D., D. Pepyne, B. Philips, and Coauthors, 2009: Short-Wavelength Technology and the Potential For Distributed Networks of Small Radar Systems. Bull. Amer. Meteor. Soc., 90, 1797-1817.

7. Page 4, Line 82: ...normalized standard error (NSE)... NSE has been expanded too many times all through this manuscript. Please pay attention to the usage of acronyms.

Thank you for this comment. We have fixed these along with the mean absolute error expansions.

8. Page 4, Line 84: ... relatively few articles on X-band... There are many X-band QPE studies. Please rephrase the writing. Again, the uniqueness of this study should be emphasized (particularly the local precipitation microphysics).

We appreciate this comment and have made the necessary change.

9. Page 4, Line 93: Over 100 different algorithms were implemented... This is very strange! Do you really need 100+ algorithms? The authors should pay more attention to the algorithms that can better reflect local rainfall microphysics. Most of the relations are taken from other papers that focused on different regions. Some of the algorithms used in this study were not even designed for X-band... A detailed investigation of local precipitation microphysics will be helpful.

Thank you for this comment. For future studies where quality control and calibration have become more concrete, research into adding more gauges for ground-truth as well as disdrometic data will be available and published.

10. Page 5, Lines 105-106: ...X-band radars will allow further indications as to whether they should be installed in regions devoid of optimal NWS WSR-88D coverage. The description on such aspect is weak. The authors may want to rephrase this sentence or refer to other studies.

We appreciate this comment, and have altered the text to

11. Page 7, Lines 162-163: R and Z ... should be independent of radar wavelength. This is true only when you are assuming Rayleigh scattering. Please clarify!

We greatly appreciate this comment and have addressed this issue accordingly.

12. Page 9, Line 210: $p < 0.10$ What is p? Probability?

We have indicated that p is, indeed, probability. Thank you for addressing this.

13. Page 23, Figure 2. Please include corresponding rainfall products derived from this radar.

Please accept our apologies, but Figure 2 corresponds to the gauge accumulated precipitation amount measured at each location. Additionally, no rainfall products / algorithms were derived specifically for this radar.